# Formulation, Optimization, and Evaluation of Transferosomes Co-Loaded with Methotrexate and Sorafenib for Anti-Arthritic Activity

**DOI:** 10.3390/pharmaceutics17091196

**Published:** 2025-09-15

**Authors:** Muhammad Adnan, Lateef Ahmad, Muhammad Junaid Dar, Humzah Jamshaid, Muhammad Noman, Muhammad Faheem

**Affiliations:** 1Department of Pharmacy, University of Swabi, Swabi 23561, Pakistan; m_adnan14@yahoo.com (M.A.); faheem@uoswabi.edu.pk (M.F.); 2Faculty of Pharmacy, MY University, Islamabad 44000, Pakistan; muhammad.nomankhan11@gmail.com; 3Department of Pharmacy, Ibadat International University, Islamabad 45750, Pakistan; muhammad.junaid@pharm.uol.edu.pk (M.J.D.); humzahjamshaid@gmail.com (H.J.)

**Keywords:** transferosomes, methotrexate, sorafenib, anti-arthritic activity

## Abstract

**Purpose:** This study was designed to develop a nanoparticle-based methotrexate (MTX) and sorafenib (SRF)-loaded transferosome (MTX-SRF-TFS) for effective management of arthritis through the transdermal route. **Methods:** For the preparation of MTX-SRF-TFS, the thin-film hydration technique was selected and optimized using Box–Behnken Design. The particle size of the nanoparticles was determined using a Malvern Zeta sizer and electron microscopy. An in vivo skin retention and penetration study was also conducted to evaluate the designed delivery system. Furthermore, the therapeutic response of MTX-SRF-TFS was determined using the CFA-induced mouse model. **Results:** The optimized MTX-SRF-TFS formulation (F4), having an average particle size (PS) of 162.20 ± 2.89 nm and percent entrapment efficiency (%EE) of MTX and SRF of 92.16 ± 4.95 and 81.54 ± 3.23, respectively, was selected for further assessment. Due to the deformable nature of MTX-SRF-TFS, MTX and SRF penetrate more deeply into the cutaneous layers, exhibiting an enhanced transdermal effect, as shown by the results of ex vivo skin permeation and retention studies. Furthermore, in vivo anti-arthritic studies have shown the superior pharmacodynamic response of MTX and SRF when incorporated into transferosomes, as it caused a marked reduction in arthritic score and paw diameter in CFA-induced arthritis in BALB/c mice. Histopathology analysis and X-ray radiography also confirmed the findings that MTX-SRF-TFS has improved anti-arthritic response in contrast to plain MTX-SRF gel. **Conclusions:** The MTX-SRF-TFS is highly effective in managing CFA-induced arthritis, and the designed delivery system should be further evaluated on pharmacokinetic grounds to progress towards clinical studies.

## 1. Introduction

Rheumatoid arthritis (RA) is an autoimmune, chronic inflammatory disorder. It is primarily associated with affecting the small joints symmetrically, having the potential of progressive bone destruction, joint damage, and synovial hyperplasia along with severe pain and joint stiffness [1]. It has marked socioeconomic significance owing to its association with physical impairment, thus compromising the mobility and quality of life of the patient [2]. The accelerated infiltration of synovial space with macrophages, neutrophils, chondrocytes, and fibroblasts, which are the production house for pro-inflammatory mediators like tumor necrosis factor-α (TNF-α), reactive oxygen species (ROS), prostaglandins, interleukins, proteases, and vasoactive amines, is responsible for RA [3,4,5].

Methotrexate (MTX), a dihydrofolate reductase inhibitor from the class of disease-modifying anti-rheumatic drugs (DMARDs), has been utilized extensively as an anchor drug to manage the symptoms associated with RA by alleviating joint damage and disease progression [6]. Having a LogP of −1.8, MTX has poor aqueous solubility, which leads to inadequate permeability after oral administration. This issue leads to low bioavailability and a short biological half-life. Another aspect is that MTX also manifests high nephro- and hepatotoxicity, which limits its clinical use [7].

Sorafenib (SRF), a multikinase inhibitor drug used to manage advanced liver cancer, was also found to be effective against arthritis in an animal model, as it reduces the angiogenesis required for proliferation of inflammatory cells and reduces tissue vascular endothelial growth factor (VEGF) levels [8,9]. SRF also has poor aqueous solubility, which reduces its chance of making it a suitable candidate for transdermal drug delivery [10].

The transdermal route is a suitable alternative to obtain higher drug bioavailability at the diseased site with reduced systemic side effects [11]. Previously, it has been reported that the incorporation of drugs inside the nano-drug delivery system (nano-DDS) minimizes the pharmacokinetic impediments of the drugs as well as the associated systemic side effects [12]. In this study, the Transfersomes (TFS) were selected as a suitable nanocarrier for the transdermal delivery of both MTX and SRF anti-arthritic agents because they facilitate targeted delivery with minimal systemic toxicity [13]. In contrast to the conventional lipid nanoparticles, i.e., liposomes and noisomes, the deformable nature of TFS (due to the presence of an edge activator) will facilitate more efficient targeted systemic delivery of drugs via the transdermal route [14]. This study is designed to evaluate the synergistic effect of the co-loading of MTX and SRF inside the TFS vesicles for RA.

Mechanistically, methotrexate (MTX) works by inhibiting purine and pyrimidine synthesis, which in turn suppresses the proliferation of immune cells and reduces joint inflammation. Sorafenib, on the other hand, targets vascular angiogenesis at the inflamed site, thereby limiting the infiltration and movement of immune cells and inflammatory mediators into the synovial tissue. Together, these complementary mechanisms act synergistically to suppress joint inflammation in arthritis [9,15].

The MTX-SRF-TFS formulations were optimized using Design Expert software, Version 9. The fabricated nano-DDS was also characterized to evaluate its nano-sized properties and drug entrapment, as well as the ex vivo skin permeation and retention potential. Furthermore, the therapeutic response of MTX-SRF-TFS was evaluated using the CFA-induced arthritic model, and the response was noted by using physical examination, histopathology analysis, and X-ray radiographs.

## 2. Materials and Methods

### 2.1. Materials

MTX, SRF, and Tween 80 were purchased from Sigma Aldrich (Gillingham, UK). Phospholipon 90G was a kind research gift from the Lipoid GMBH (Ludwigshafen, Germany). Carbopol, Sodium Chloride, Chloroform, Methanol, Disodium Hydrogen Phosphate, and Dipotassium Hydrogen Phosphate were purchased from Duksan Chemicals (Ansan, Republic of Korea).

### 2.2. Animals

To perform an ex vivo skin permeation study and an in vivo antiarthritic study, 30 BALB/c mice were provided by Riphah International University, Islamabad. Animals were retained in steel cages and acclimatized in the animal house. All animals were fed a standard animal diet, clean water was provided, and they were kept under a 12/12-h light–dark cycle.

### 2.3. Preparation of MTX-SRF-TFS

The MTX-SRF-TFS were formulated by the thin-film hydration technique as previously reported [16]. The organic phase was prepared by dissolving the Phospholipon 90G (P-90G), Tween 80, and SRF in an equimolar (1:1) mixture of methanol and chloroform, which was then consigned to a rotary evaporator operated under reduced pressure, having the temperature of 60 °C and rotation of 100 rpm for the formation of a thin film after the evaporation of the organic solvent. After that, the aqueous phase was fabricated by mixing the Tween 80 and MTX in phosphate buffer pH (7.4), which then hydrated the thin film at a temperature for 1.0 h and was then subjected to extrusion through the polycarbonate syringe filters of 0.45 µm for 3 times, followed by 2 times from 0.22 µm, to downsize the formulated TFS [17].

### 2.4. Optimization of SRF-MTX-TFS

Box–Behnken Design from the Design Expert^®^ (version 12 Stat Ease) was employed to optimize the formulation, and the selected dependent and independent variables are stated in Table 1. All 13 formulations given by the design expert were formulated, and the effect of independent factors (P-90G, Tween 80, and MTX: SRF) was analyzed over dependent factors (Particle size and %EE of MTX and SRF) [14].

### 2.5. Particle Size, Zeta Potential, Polydispersability Index, and External Morphology of SRF-MTX-TFS

The fabricated SRF-MTX-TFS were analyzed for various physicochemical factors, including particle size (PS), zeta potential (ZP), and Polydispersability Index (PDI), using the Zetasizer (ZS-90, Malvern equipment, Worcestershire, UK) at 25 °C. The sample was diluted with distilled water before evaluation. TEM analysis of the optimized formulation was executed to confirm the surface morphology, size, and shape by Transmission Electron Microscopy (Hitachi High Tech Ltd., Tokyo, Japan). Prior to evaluation, the sample was stained negatively by utilizing phosphotungstic acid, and the examination was performed at a voltage of 120 kV. The MTX-SRF-TFS dispersion was freeze-dried in a lyophilizer (Christ Alpha 1-2 LDplus, Osterode, Germany) in order to generate nanoparticle pellets for TEM analysis [18,19].

### 2.6. SRF and MTX Entrapment Efficiency of SRF-MTX-TFS

The indirect method was used to determine the entrapment efficiency of MTX-SRF-TFS. For this, fabricated TFS were centrifuged for 3 h at 15,000 rpm to separate the free drug using the centrifuge machine (Z-32HK, HERMLE) [20]., Gosheim, Germany) [21]. After that, the supernatant was separated and analyzed using a spectrophotometer at 303 nm and 264 nm for the quantification of MTX and SRF, respectively.

The following mathematical equation was used to compute the %EE.(1)%EE=B1B2×100
where B1 is the quantity of drug entrapped and B2 is the quantity of total drug added.

### 2.7. Designing and Assessing of SRF-MTX-TFS Gel

To make the SRF-MTX-TFS gel rheologically suitable for topical application, Carbopol gel was chosen as the final dosage form. To prepare the gel, a 1% *w*/*v* carbopol solution was required, which was prepared by adding 100 mg carbopol 934 in 10 mL distilled water with continuous stirring for 3 to 4 h until the gel was formed, and then the formulated MTX-SRF-TFS was added succinctly. Finally, a few drops of triethanolamine were added to neutralize the dispersion. The synthesized gel was examined for its physical appearance, pH, homogeneity, and color. The sample for pH measurement was prepared by adding 1 g of gel to 40–50 mL of distilled water, and the pH was determined using a pH meter, and the readings were taken in triplicate to avoid the probability of error.

### 2.8. Skin Permeation and Deposition Studies

A Franz diffusion cell having the donor and acceptor compartments (having a capacity of almost 12 mL) was employed to account for the permeation and flux of the formulation and its comparison with the formulation-loaded gel and blank gel. Briefly, to obtain the skin, BALB/c mice were euthanized, the skin was cut and shaped properly, and fat was removed and placed on the diffusion cell so that the stratum corneum (SC) layer of the skin was headed outwards. The MTX-SRF-TFS, MTX-SRF-TFS gel, and plain MTX-SRF gel, having a concentration of 1 mg, were stationed in the donor, and phosphate buffer (PBS), having a pH of 7.4, was placed in the acceptor compartment. The temperature was maintained at 32 ± 1 °C, and this study was performed for 24 h. Samples of 0.5 mL were collected at predetermined intervals of 0.25, 0.5, 1, 2, 4, 8, 12, and 24 h, and the PBS was replaced with the same volume at each interval in the receiver compartment [21]. The withdrawn samples were analyzed under a UV spectrophotometer (Model: Cary 60, Agilent Technologies, Penang, Malaysia) for quantification, and the graphs were plotted against the cumulative amount of drug permeated and sampling time. Moreover, the enhancement ratio (ER) was calculated utilizing the following equation:(2)$$ER=\frac{J_{max}\;of\;MTX-SRF-TFS}{J_{max}\;of\;blank\;gel}$$

The skin tissue from the cells was removed, the surface was cleaned, and SC was removed using 10 cellophane adhesive tape strips. Following this, the tape strips were soaked in the extraction buffer with 70% methanol and 30% ammonium acetate. The stripped skin tissue was then chopped into minute slices and subjected to soaking in the same buffer (for 1 h), followed by probe sonication. Finally, after extrusion with a 0.4 micron syringe filter, the sample will be analyzed for drug quantification using a UV spectrophotometer at 303 nm and 264 nm for the quantification of MTX and SRF, respectively [16].

### 2.9. In Vivo Antiarthritic Activity

The antiarthritic response of MTX-SRF-TFS was determined in a Complete Freud’s Agent (CFA)-induced arthritic model in BALB/c mice. In this study, 25 μL of CFA was injected intradermally into the animal’s footpad in all animal groups except the control group. Moreover, 25 μL of normal saline was injected into the control group. On day 14 of the CFA injection, the animals with an arthritic score of 3–4 were further recruited for this study, and treatment was initiated. The animals were divided into four experimental groups, including

➢Negative Control (immunized with CFA and kept untreated).➢Normal group (not injected with CFA and treated with normal saline).➢Plain MTX-SRF gel (injected with CFA, followed by the treatment) (MTX dose 0.5 mg/kg/day and SRF 10 mg/kg/day).➢MTX-SRF-TFS gel (injected with CFA followed by the treatment) (MTX dose 0.5 mg/kg/day and SRF 1.75 mg/kg/day).

In the plain MTX-SRF gel, each animal received a daily dose equivalent to 0.5 mg/kg of MTX and 10 mg/kg of SRF. In contrast, the MTX-SRF-TFSG formulation delivered the same MTX dose (0.5 mg/kg) but with a considerably lower SRF dose (1.75 mg/kg). Previous studies have demonstrated that SRF, at a dose of 10 mg/kg in nanoparticle form, exhibits significant anti-arthritic activity. The formulation was administered in gel form and applied topically to the paw of mice. The rationale for using a reduced SRF dose in the current formulation lies in the synergistic interaction between SRF and MTX, which enhances therapeutic efficacy even at lower concentrations.

The treatment was continued for 2 weeks, and the arthritic scoring and paw thickness were determined on days 14, 16, 20, 24, and 28 to evaluate the treatment response in all the experimental groups, followed by the histopathology analysis and X-ray of the paw joints [7,18,22].

### 2.10. Radiology Studies

The excised paw joints of the 28th day of this study from the animals of all groups were subjected to X-ray examination (Leonardo DR mini II, OR Technology (Oehm und Rehbein GmbH, Rostock, Germany) to evaluate the bone and cartilage erosion as well as the degree of swelling [7].

### 2.11. Animal Joint Tissue Histology

Animals were sacrificed on the 28th day of this study, and the right paw joint was excised. These joint tissues were stored in 4% formaldehyde buffer for 48 h. Decalcification of the joint was performed by incubating the tissue with 5% formic acid. The decalcified tissue was then embedded in paraffin, and the cross sections of 5 μm tissue were made using a cryostat and stained with hematoxylin and eosin (H&E). Finally, the compound microscope (Olympus, West Midlands, UK) was employed to take the microphotographs [7].

### 2.12. Statistical Analysis

In this study, the formulations were optimized using the Design Expert^®^ software, Version 9.Furthermore, all the values are mentioned as mean ± SEM (*n* = 3, except for the in vivo anti-arthritic study, where *n* = 6). One-way ANOVA and multiple comparison Tukey’s tests were applied using SPSS (ver. 30) and GraphPad Prism^®^ (ver. 10.4.2).

## 3. Results and Discussion

### 3.1. Box–Behnken Design for SRF-MTX-TFS Optimization

The Box–Behnken experimental design has provided 13 experiments that assessed the influence of independent variables, including P-90G, Tween 80 concentration, and the drug ratio, on the particle size of SRF-MTX-TFS vesicles as well as the %EE of SRF and MTX inside the SRF-MTX-TFS. All the experimental runs and their determined response factors are stated in Table 2. Moreover, the detailed parameters of the Box–Behnken Design and the design expert-generated 3D graphs are shown in Table 3 and Figure 1. As a result, the F4 experimental run was considered an optimized formulation because all the response factors were in the desired range.

#### 3.1.1. Effect of Independent Factors on the SRF-MTX-TFS Particle Size

The mean particle size distribution of all 13 formulations (runs generated by Design Expert) was found to be in the range of 131.30 ± 2.87 nm to 267.44 ± 6.78 nm. The P-90G (A_1_) and Tween 80 (A_2_) concentrations have a significant effect on the particle size of SRF-MTX-TFS, with *p*-values of 0.007 and 0.044, respectively. From the 3D graph presented in Figure 1, it can be observed that the size of the SRF-MTX-TFS vesicle increases with an increment in P-90G concentration, as F7 demonstrated with the size of 131.30 ± 2.87 nm and F11 with the vesicle size of 267.44 ± 6.78 nm because in the formulation of F7 and F11, 100 mg and 200 mg of P-90G were utilized. A similar effect is also evident from the particle size results of F1 and F13. On the other hand, Tween 80 has an opposite effect on the particle size of the SRF-MTX-TFS, as can be observed from the 3D graph illustrated in Figure 1; the large-sized particles were formed at a low level of Tween 80 and vice versa. By keeping the P-90G concentration and drug ratio, the size of the vesicles increased incrementally from 204.29 ± 3.22 nm to 267.44 ± 6.78 nm when the Tween 80 concentration was increased from 5% *w*/*w* to 20% *w*/*w* (Table 2). In a similar way, F2 and F7 also demonstrated the above-stated effect on the particle size by altering the concentration of Tween 80. Drug ratio has a non-significant effect on the particle size of the SRF-MTX-TFS (as *p*-value), which is evident from the 3D graph in Figure 1. This is because of the fact that with low lipid content, the unilamellar TFS were obtained [16]. However, in the case of surfactant, an increment in surfactant concentration reduces the particle size because it reduces surface tension and prevents the vesicles from agglomeration, as reported by Moolakkadath et al. [19]. Previously, it has been reported that for effective transdermal delivery, the particle size of the vesicle should be below 300 nm [23]. Thus, the mean particle size of all the designed formulations was found to be below the stated value.

#### 3.1.2. Effect of Independent Factors on the %EE of SRF

The entrapment efficiency of SRF inside SRF-MTX-TFS was also considered as a critical parameter for the optimization because its therapeutic efficacy is directly dependent upon the drug entrapped inside the vesicles. From Figure 1, the 3D graph shows the fact that by keeping the drug ratio constant, the %EE of SRF increases with an increment in the concentration of both Tween 80 and P-90G. The %EE of SRF of F2 and F11 was 52.93 ± 2.01 and 72.33 ± 2.72, respectively. This increment was mainly due to higher P-90G content in F11. In the SRF-MTX-TFS, the entrapment of SRF occurs in the lipid bilayer. Hence, with higher lipid content, its entrapment surged. Furthermore, Tween 80 also affects this parameter with the pretext that it facilitates the solubilization of lipophilic drugs like SRF [24]. This effect can also be observed by comparing the %EE of F2 and F7. Moving from a low level of Tween 80 to its high level, the %EE of SRF inside SRF-MTX-TFS rose from 52.93 ± 2.01 to 82.39 ± 1.72, as stated in Table 2. The drug ratio is also significantly influencing the %EE of SRF. The higher the concentration of SRF, the more entrapment of SRF there is. By comparing the %EE of SRF of F1 and F3, it was observed that by increasing the SRF quantity, its %EE also increased from 68.44 ± 2.17 to 82.16 ± 4.88.

#### 3.1.3. Effect of Independent Factors on the %EE of MTX

MTX entrapment was also significantly influenced by P-90G, Tween 80, and drug concentration utilized for the synthesis of SRF-MTX-TFS, as their *p*-values were 0.004, 0.009, and 0.04. As a BCS class III drug, the MTX is more hydrophilic and entrapped inside the core of the nano-vesicle. Therefore, with a high level of P-90G, the entrapment of MTX was also elevated because large-sized particles were formed. This can be observed in Figure 1 and Table 2. By evaluating the MTX %EE of F1 and F13, it can be assessed that if the level of Tween 80 and drug ratio remained constant, the MTX %EE would increase from 69.54 ± 1.84 to 89.56 ± 2.82,with an elevation of P-90G conc. from 100 mg to 200 mg. As far as Tween 80 concentration is considered, the MTX %EE was reduced significantly with an elevation of Tween 80 concentration because in higher concentrations, Tween 80 induces pore formation in the lipid bilayer of TFS, which ultimately causes leakage of entrapped drug. With an increment in surfactant concentration, the entrapment of the hydrophilic drug reduces; this finding was also in accordance with the results of already published work [19]. As observed with F11 and F12 (Table 2), the MTX %EE was higher with a lower level of Tween 80 concentration, i.e., 5% *w*/*w*. A similar pattern was also seen with F2 and F7.

### 3.2. Zeta Potential and Polydispersability Index of Optimized SRF-MTX-TFS

The particle size was considered as a response factor for the optimization of SRF-MTX-TFS. However, the polydispersability and zeta potential were only assessed for F4 (optimized SRF-MTX-TFS). Having a particle size of 162.20 ± 2.80 nm, the polydispersability and zeta potential were found to be 0.170 ± 0.005 and −31.6 ± 1.0 mV, respectively, as displayed in Figure 2A,B. Results predicted the superior stability of SRF-MTX-TFS, since the higher average zeta potential of SRF-MTX-TFS prevented the aggregation of nanoparticles [25].

### 3.3. External Morphology of Optimized SRF-MTX-TFS

Results of TEM images, illustrated in Figure 2C, have shown significant correlation with the DLS results of the optimized MTX-SRF-TFS. The image also demonstrates the spherical morphology of vesicles with a clear, intact, and smooth surface. The TEM micrograph (illustrated in Figure 2C) also confirms the mean particle size distribution results obtained through DLS (Malvern Zeta Sizer).

### 3.4. Ex Vivo Permeation and Retention Assay

The aim of selecting transferosomes as a drug delivery system is that it exhibits potential transdermal flux, usually barred by a tightly packed SC skin layer. The presence of edge activators, as a major ingredient, imparts deformable and reformable characteristics to the transferosomal vesicle. According to the graph illustrated in Figure 2D,E, encapsulation of SRF and MTX inside the TFS vesicles significantly increased the transdermal flux. The plain MTX-SRF was presented with 54.29 ± 13.53 µg/cm^2^ of MTX transdermal flux in 24 h. This was significantly enhanced with MTX-SRF-TFS and MTX-SRF-TFS gel and shown to have transdermal flux of 403.81 ± 23.51 and 287.12 ± 12.16 µg/cm^2^, respectively. Similarly, in the case of SRF, the cumulative amount of SRF permeated through mounted skin was also raised markedly with MTX-SRF-TFS and MTX-SRF-TFS gel, as illustrated in the graph. From the outset, the SRF release from SRF-MTX-TFS increased significantly and presented with 51.65 ± 17.21 µg/cm^2^ of transdermal flux in just 4 h. It surged to 517.15 ± 32.55 µg/cm^2^ in 24 h. The permeation of MTX and SRF from MTX-SRF-TFS gel, 287.12 ± 26.12 and 342.12 ± 53.93 µg/cm^2^, is non-significantly less than MTX-SRF-TFS mainly due to the release of retardant potential of carbopol gel (Table 4). The skin retention of both drugs was also evaluated to compare the superiority of the TFS system in facilitating the transdermal drug delivery. From the graph presented in Figure 2F,G, it can be observed that there was superficial drug retention with plain MTX-SRF-TFS gel, mainly in the stratum corneum (5.06 ± 1.11%). However, with MTX-SRF-TFS and MTX-SRF-TFS gel, both drugs can penetrate deeper cutaneous layers (deeper epidermis and dermis) in a better way, leading to significantly enhanced transdermal flux. After incorporation inside TFS, the cutaneous permeation of MTX and SRF becomes higher than the cutaneous retention because the deformable nature of the vesicles facilitates the drugs crossing the skin tissue and reaching the systemic circulation [18].

### 3.5. In Vivo Anti-Arthritic Study: Arthritic Score and Paw Thickness

As discussed previously, CFA was employed to induce arthritis in BALB/c mice, and upon injecting CFA in the sub-plantar region of the mouse paw, the symptoms (swelling and redness) were pronounced and became maximum on day 14, with severe swelling and widespread erythema. The animals having a 3–4 arthritic score were allocated for further experimentation. As seen in Figure 3A,B, the animals in the negative group were not given any treatment; hence, their arthritic score remained 3.7 ± 0.6 (Figure 3B), which predicts severe arthritic symptoms, viz., intense swelling and inflammation. Furthermore, BALB/c mice paw thickness also exhibits similar findings (Figure 3C). The arthritic score in the plain MTX-SRF-gel and MTX-SRF-TFSG groups was reduced to 2.7 ± 0.6 and 1.33 ± 0.5, respectively, after 14 days of treatment. Results predict that the disease symptoms are significantly (*p* < 0.001) reduced with MTX-SRF-TFSG. However, the plain MTX-SRF gel leads to a non-significant reduction in arthritic symptoms.

The treatment was initiated on day 14, and with the application of plain MTX-SRF gel, the animal’s paw diameter reduction was observed. From day 14 to day 28, the animal paw diameter was non-significantly reduced from 2.67 ± 0.02 mm to 2.25 ± 0.09 mm, as shown in Figure 3A,C. In contrast, the treatment with MTX-SRF-TFS gel has significantly lowered the animal paw swelling from 2.56 ± 0.14 mm (day 14) to 1.92 ± 0.07 mm (day 28). As shown in the paw photographs in Figure 3A, the findings depicted that the MTX-SRF-TFS gel has significantly (*p* < 0.0001) reduced the arthritic symptoms, as evident by the paw diameter, which is comparable to the normal group. Adequate transdermal penetration and distribution of MTX and SRF at the inflamed area (passive targeting) would be the pretext for a superior therapeutic response [18].

### 3.6. Effect on BALB/c Mice Body Weight

In a diseased state, the body weight of the animal often declines, which is an indicator of illness and can be used to track the response to the treatment. As illustrated in the graph shown in Figure 3D, the mice’s body weight declined from 25.28 ± 0.47 g to 20.91 ± 0.89 g from day 14 to day 28 in the negative control group. However, no weight reduction was observed in the normal group. Furthermore, the treatment of arthritic mice with plain MTX-SRF-TFSG gel has displayed a significant reversal of weight loss in addition to the non-significant (*p* > 0.05) weight variation in contrast to the normal group [26].

### 3.7. X-Ray Radiographs

To further assess the cartilage and bone damage, paw joint X-ray radiology has been conducted. As illustrated in Figure 4(AII), the negative control group exhibits a higher degree of joint inflammation along with bone damage. Treatment with plain MTX-SRF gel has reduced the joint inflammation, but not very effectively. However, the MTX-SRF-TFS gel was associated with a marked reduction in joint inflammation and bone erosion, illustrated in Figure 4(AVII,AVIII). The superior response to MTX-SRF-TFS gel was believed to be due to higher drug distribution to the inflamed area, and the results of X-ray radiology are in accordance with the results of histopathology analysis [7].

### 3.8. Bone and Cartilage Erosion

Arthritis is an inflammatory joint disorder accompanied by marked bone and cartilage erosion and hyperinflammation of synovium, as well as joint space narrowing [27]. Joint histopathology analysis has been conducted to evaluate the response to treatment. According to the microphotographs illustrated in Figure 4(BIX), the negative control group displayed a marked reduction in joint gaps followed by synovium inflammation along with cartilage damage as indicated by the yellow arrow. These findings were absent in the normal group, i.e., having intact synovium and adequate joint gap. As far as treatment groups are concerned, the application of MTX-SRF-TFSG has significantly healed the eroded synovium and the cartilage, as well as reduced the synovium (Figure 4(BXII)) [7].

## 4. Conclusions

In this study, the MTX-SRF-TFS has successfully been designed using the thin-film hydration method and optimized based on average particle size and %EE of MTX and SRF. F4, the optimized formulation, showed to have improved cutaneous permeation and some retention in the deeper cutaneous layer, which could be due to its size and deformable nature. Finally, the MTX-SRF-TFS gel was prepared and evaluated for its antiarthritic response in CFA-induced arthritis in BALB/c mice. Results have shown an increased anti-arthritic response of MTX-SRF-TFS, which demonstrated a good arthritic score, reduction in paw thickness, a sufficiently intact synovial membrane, and a decrease in inflammatory cells. X-ray radiographs also supported these findings.

## Figures and Tables

**Figure 1 pharmaceutics-17-01196-f001:**
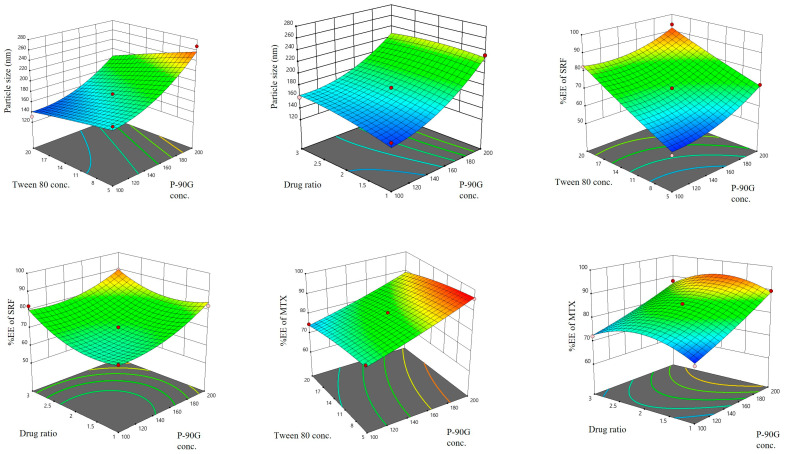
Three-dimensional curves illustrating the effect of the independent variable on the dependent factors.

**Figure 2 pharmaceutics-17-01196-f002:**
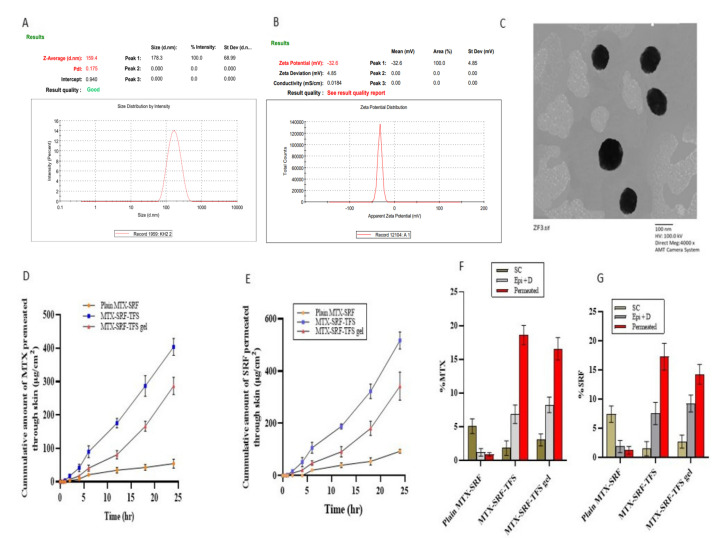
(**A**) Mean particle size distribution of MTX-SRF-TFS; (**B**) mean zeta potential of MTX-SRF-TFS; (**C**) TEM microphotograph of optimized MTX-SRF-TFS; (**D**,**E**) MTX and SRF permeated across the skin; (**F**,**G**) percent of MTX and SRF retained and permeated across the skin.

**Figure 3 pharmaceutics-17-01196-f003:**
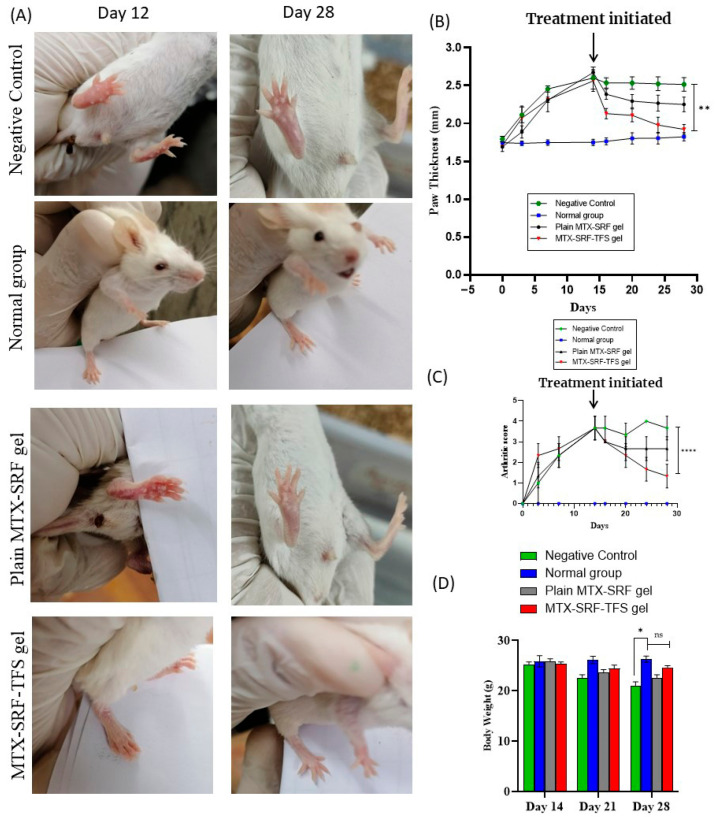
(**A**) BALB/c mice photographs of an in vivo anti-arthritic study illustrating the superior therapeutic response of MTX-SRF-TFS gel. (**B**) Arthritic score of BALB/c mice during in vivo anti-arthritic study; (**C**) paw thickness in millimeters of BALB/c mice during in vivo anti-arthritic study; (**D**) body weight variation of BALB/c mice during in vivo anti-arthritic study. All the values are mentioned as mean ± S.D. with (*n* = 6). (ns Non-significant, * *p* < 0.005, ** *p* < 0.0001, **** *p* < 0.00001).

**Figure 4 pharmaceutics-17-01196-f004:**
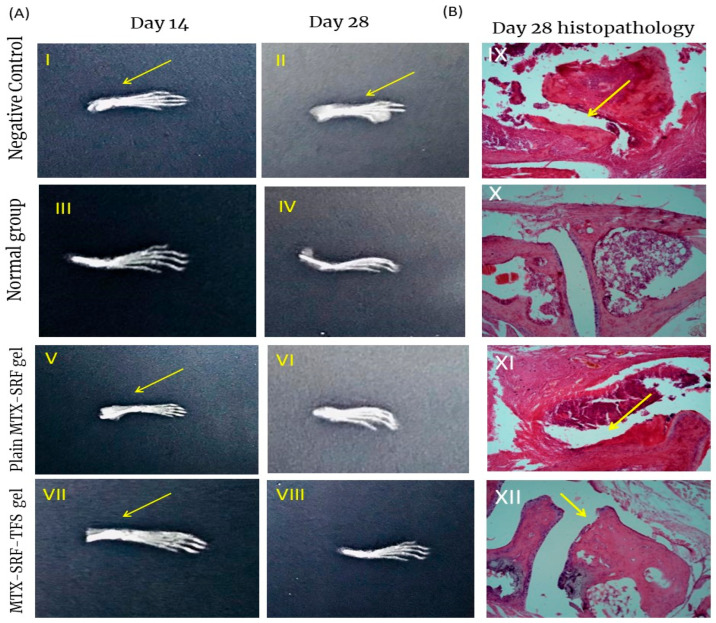
(**A**) X-ray radiographs of BALB/c mice at day 14 and day 28 of the in vivo anti-arthritic, study, arrows in the X-ray indicates inflammation in synovial joint (**B**) Histopathological alterations in the paw joint of BALB/c mice, arrows demonstrates disruption/inflammation in synovial membrane in paw joint.

**Table 1 pharmaceutics-17-01196-t001:** Independent and dependent factors selected for optimization of SRF-MTX-TFS using Box–Behnken Design.

Independent Variables	Levels Assessed		
	Low Level (−1)	Medium Level (0)	High Level (+1)
A1: Lipid Conc. (mg)	100	150	200
A2: Tween80 Conc. (% *w*/*w*)	5	12.5	20
A3: Drug Ratio	1:1	1:2	2:1
**Dependent Factors**	**Required response**		
B1: Particle Size (nm)	Minimize		
B2: Entrapment Efficiency of SRF (%)	Maximize		
B3: Entrapment Efficiency MTX (%)	Maximize		

**Table 2 pharmaceutics-17-01196-t002:** Optimization of formulation of SRF-MTX-TFS using independent variables and dependent factors.

Formulation	Independent Variables			Dependent Factors		
	A_1_	A_2_	A_3_	B_1_	B_2_	B_3_
F1	100	12.5	1:1	145.70 ± 2.61	68.44 ± 2.17	69.54 ± 1.84
F2	100	5	1:2	174.76 ± 1.87	52.93 ± 2.01	79.68 ± 2.06
F3	100	12.5	2:1	159.44 ± 3.89	82.16 ± 4.88	72.12 ± 3.21
F4	150	20	2:1	162.20 ± 2.80	92.16 ± 4.95	81.54 ± 3.23
F5	150	20	1:1	139.64 ± 1.56	84.27 ± 3.67	73.92 ± 2.65
F6	200	12.5	2:1	220.02 ± 6.54	89.62 ± 3.33	87.89 ± 4.44
F7	100	20	1:2	131.30 ± 2.87	82.39 ± 1.72	74.73 ± 1.76
F8	150	5	1:1	179.05 ± 3.65	62.76 ± 2.13	82.67 ± 5.24
F9	150	5	2:1	185.44 ± 3.65	68.32 ± 3.21	78.78 ± 3.38
F10	150	12.5	1:2	176.27 ± 4.13	70.45 ± 5.92	84.95 ± 2.99
F11	200	5	1:2	267.44 ± 6.78	72.33 ± 2.72	95.75 ± 5.64
F12	200	20	1:2	204.29 ± 3.22	93.84 ± 5.04	88.45 ± 3.05
F13	200	12.5	1:1	231.55 ± 4.98	82.32 ± 3.16	89.56 ± 2.82

**Table 3 pharmaceutics-17-01196-t003:** Summary of the applied Box–Behnken Design for SRF-MTX-TFS.

Dependent Responses	R^2^	Adjusted R^2^	Significant Factors	S. D	Adequate Precision	Expected Optimized Parameters	Actual Optimized Parameters
Particle Size (nm)	0.9538	0.8150	A_1_, A_2_	16.55	8.303	161.77	159.40
EE% of SRF	0.9876	0.9503	A_1_, A_2_, A_3_	2.73	16.262	85.60	92.16
EE% of MTX	0.9928	0.9713	A_1_, A_2_	1.33	22.130	84.17	81.54

**Table 4 pharmaceutics-17-01196-t004:** Ex vivo permeation data of plain MTX-SRF-gel, MTX-SRF-TFS, and MTX-SRF-TFS gel.

Formulation	Total Drug Amount Permeated in 24 h Q (μg/cm^2^)		Flux J_max_ (μg/cm^2^/h)		Enhancement Ratio	
	SRF	MTX	SRF	MTX	SRF	MTX
MTX-SRFgel	92.84 ± 7.98	54.29 ± 13.53	3.86 ± 0.33	2.27 ± 0.57	1	1
MTX-SRF-TFS	517.15 ± 32.56	403.81 ± 25.5	21.46 ± 1.35	16.73 ± 0.96	5.55	7.37
MTX-SRF-TFS gel	342.12 ± 53.93	287.12 ± 26.12	14.25 ±2.24	11.92 ± 1.15	3.69	5.25

## Data Availability

Data will be made available upon request.

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
