# Peer review of "Formulation, Optimization, and Evaluation of Transferosomes Co-Loaded with Methotrexate and Sorafenib for Anti-Arthritic Activity"

_pharmaceutics, 2025, doi:10.3390/pharmaceutics17091196_

Round 1
Reviewer 1 Report
Comments and Suggestions for Authors
In this study, methotrexate and sorafenib were loaded onto deformable transfersomes by the thin-film hydration method to prepare a transdermal drug MTX-SRF-TFS with uniform particle size, high skin permeability and retention, and good therapeutic effect in a mouse arthritis model.
The manuscript is well-organized and has high research value. After minor revisions, it can be considered for publication. The following issues can be optimized and modified:
1)In "2.3. Preparation of MTX-SRF-TFS", the words "Initially" and "Erstwhile" are inappropriately used in the description of the preparation process, resulting in logical errors in the sentence, which should be corrected.
2)According to the description of the preparation process of MTX-SRF-TFS, the obtained product is nanoparticles dispersed in a liquid. How is TEM analysis conducted in this state? A detailed description of the sample preparation process for TEM analysis of MTX-SRF-TFS should be provided.
3)In "2.7. Designing and assessment of SRF-MTX-TFS loaded Carbopol gel", the abbreviation is SRF-MTX-TFS loaded carbopol gel (SRF-MTX-TFSG), while in Table 4 and Figure 3, it is referred to as SRF-MTX-TFS gel. The abbreviation should be consistent.
4)In "2.9. In vivo antiarthritic activity", there are errors in the names corresponding to the groups after grouping. "Immunized with CFA and kept untreated" is the model control group rather than the Negative Control, and "not injected with CFA and treated with normal saline" is the blank group or negative control, not the positive control. Additionally, the author's setting of the control groups does not match the names in Figure 3.
5)In "2.9. In vivo antiarthritic activity", the form of administration, site and dosage of the drug should be described in detail.
Author Response
We thank the reviewers for their critical evaluation of our manuscript. We have now incorporated all the suggestions in the revised manuscript. The reviewers’ recommendations and comments were very helpful in improving the quality of our research manuscript. All the points raised by the reviewers have been addressed individually in the manuscript, and all the changes have been highlighted in the revised manuscript by using colored text (yellow color). We hope that these changes will make the points clear to the readers and improve the overall presentation of our manuscript.
Reviewer 1
In this study, methotrexate and sorafenib were loaded onto deformable transferosomes by the thin-film hydration method to prepare a transdermal drug MTX-SRF-TFS with uniform particle size, high skin permeability and retention, and good therapeutic effect in a mouse arthritis model. The manuscript is well-organized and has high research value. After minor revisions, it can be considered for publication. The following issues can be optimized and modified:
Comment 1: In "2.3. Preparation of MTX-SRF-TFS", the words "Initially" and "Erstwhile" are inappropriately used in the description of the preparation process, resulting in logical errors in the sentence, which should be corrected.
Response 1:
We are highly thankful to the reviewer for pointing out this in the manuscript. As per the reviewers' comments, we have modified the mentioned words, and the language has been revised throughout the article.
Comment 2: According to the description of the preparation process of MTX-SRF-TFS, the obtained product is nanoparticles dispersed in a liquid. How is TEM analysis conducted in this state? A detailed description of the sample preparation process for TEM analysis of MTX-SRF-TFS should be provided.
Response 2:
Thanks for your valuable comment. In our study, the MTX-SRF-TFS dispersion was not conducted in a liquid state; rather, this dispersion was subjected to freeze-drying (lyophilization) to obtain nanoparticles, which were subsequently characterized by TEM analysis. This has been added to the methodology of the manuscript.
Comment 3: In "2.7. Designing and assessment of SRF-MTX-TFS loaded Carbopol gel", the abbreviation is SRF-MTX-TFS loaded carbopol gel (SRF-MTX-TFSG), while in Table 4 and Figure 3, it is referred to as SRF-MTX-TFS gel. The abbreviation should be consistent.
Response 3:
We are thankful to the reviewer for pointing out this. We have revised the abbreviations. Please refer to the revised manuscript.
Comment 4: In "2.9. In vivo antiarthritic activity", there are errors in the names corresponding to the groups after grouping. "Immunized with CFA and kept untreated" is the model control group rather than the Negative Control, and "not injected with CFA and treated with normal saline" is the blank group or negative control, not the positive control. Additionally, the author's setting of the control groups does not match the names in Figure 3.
Response 4:
This is a very important aspect that will directly impact the clarity of the article. We have revised the abbreviations. Please refer to the revised manuscript.
Comment 5: In "2.9. In vivo antiarthritic activity", the form of administration, site and dosage of the drug should be described in detail.
Response 5:
Thanks for this comment; it will clarify things for the readers. The form of administration was gel, and the site of administration was the mouse paw. This has also been updated in the revised manuscript. Please refer to the section. Furthermore, for an in vivo antiarthritic study, the drugs are used in the above-mentioned doses;
For Plain MTX-SRF gel, the daily per animal dosing was comprised of 0.5 mg/kg of MTX and 10 mg/kg of SRF. However, the MTX-SRF-TFSG daily gel application provides 0.5 mg/kg of MTX and 1.75 mg/kg of SRF. In the previously published study, the SRF was used in the dose of 10 mg/kg for anti-arthritic activity in the form of nanoparticles. However, the reason of low dose of SRF is the synergism between SRF and MTX (Gözel, Çakirer et al. 2017, Wang, Liu et al. 2018, Salem, Abd El-Maboud et al. 2022).
Reviewer 2 Report
Comments and Suggestions for Authors
- Give more preclinical or clinical proof that this combination has synergistic anti-arthritic effects. In particular, how do the anti-inflammatory (MTX) and anti-angiogenic (SRF) mechanisms of folate metabolism inhibition and kinase inhibition complement one another in the pathophysiology of RA?
- Provide the complete statistical analysis, including model fit metrics other than R2
- and adjusted R2, as well as p-values for interaction terms (Table 3, lines 209–233).
- Only the optimized formulation (F4, lines 270–277) had the zeta potential and PDI reported. These characteristics should be assessed for consistency and stability across the design space for each of the 13 formulations to guarantee robustness. Are these data available from the authors?
- The spherical morphology is confirmed by the TEM study (line 282); however, no scale bar or quantitative size distribution from TEM is given. To confirm the correlation with (DLS) results, may the authors include these?
- The study shows a significant improvement in transdermal flux with MTX-SRF-TFS compared to plain gel. However, the enhancement ratio calculation compares TFS to blank gel, which should be calculated relative to plain gel.
- Recent research should be added to the reference list, along with references to stability and comparable nano-delivery methods, as well as citations demonstrating the anti-arthritic properties of SRF and the synergy between MTX and SRF. To improve the manuscript, using the following reference is recommended:”https://doi.org/10.1021/acsabm.5c00257”.
- The conclusion suggests further pharmacokinetic evaluation, but no preliminary data is provided. The authors need to provide initial pharmacokinetic data to support the claim of enhanced bioavailability. MTX and SRF are associated with significant toxicities, and any toxicity assessed in the in vivo study should be addressed.
- To highlight its distinctive feature and advantages, the text would benefit from a comparison of the MTX-SRF-TFS formulation with other transdermal nano-delivery methods (such as liposomes and ethosomes) for the treatment of RA.
Author Response
Thank you very much for taking the time to review this manuscript. Please find the detailed responses below and the corresponding revisions.
Comment 1: Give more preclinical or clinical proof that this combination has synergistic anti-arthritic effects. In particular, how do the anti-inflammatory (MTX) and anti-angiogenic (SRF) mechanisms of folate metabolism inhibition and kinase inhibition complement one another in the pathophysiology of RA?
Response 1: We are grateful to the reviewer for the clarification of this important comment. We have incorporated the following statement. This has been highlighted in the introduction.
Mechanistically, methotrexate (MTX) works by inhibiting purine and pyrimidine synthesis, which in turn suppresses the proliferation of immune cells and reduces joint inflammation. Sorafenib, on the other hand, targets vascular angiogenesis at the inflamed site, thereby limiting the infiltration and movement of immune cells and inflammatory mediators into the synovial tissue. Together, these complementary mechanisms act synergistically to suppress joint inflammation in arthritis(Gözel, Çakirer et al. 2017, Wang, Liu et al. 2018, Salem, Abd El-Maboud et al. 2022).
Comment 2: Provide the complete statistical analysis, including model fit metrics other than R2 and adjusted R2, as well as p-values for interaction terms (Table 3, lines 209–233).
Response 2: We highly appreciate your valuable comment. In our research study, we have used only these two parameters, R2 and adjusted R2, keeping the following studies as a reference. Which is already published
Development and evaluation of novel miltefosine-polyphenol co-loaded second generation nano-transfersomes for the topical treatment of cutaneous leishmaniasis (Dar, McElroy et al. 2020).
Comment 3: Only the optimized formulation (F4, lines 270–277) had the zeta potential and PDI reported. These characteristics should be assessed for consistency and stability across the design space for each of the 13 formulations to guarantee robustness. Are these data available from the authors?
Response 3:
We are highly grateful to the reviewer for raising this important aspect. We had determined these aspects (PDI and Zeta potential) only for our optimized formulation (F4) because of the limited financial resources.
Comment 4: The spherical morphology is confirmed by the TEM study (line 282); however, no scale bar or quantitative size distribution from TEM is given. To confirm the correlation with (DLS) results, may the authors include these?
Response4:
Thank you for pointing this out in the figure. As per the reviewer’s suggestion, we have incorporated the TEM micrograph with a scale bar in the revised manuscript. Please refer to Figure 2 of the revised manuscript. The scale bar shows that the particle size is in correlation with the particle size observed with DLS.
Comment 5: The study shows a significant improvement in transdermal flux with MTX-SRF-TFS compared to plain gel. However, the enhancement ratio calculation compares TFS to blank gel, which should be calculated relative to plain gel.
Response5:
We appreciate the reviewer for this comment. To clarify, the transdermal flux was evaluated by quantification of drugs penetrated and permeated from the receptor compartment to the donor compartment of the Franz diffusion cell. Therefore, we have compared the transdermal flux of MTX-SRF-TFS, MTX-SRF-TFG and MTX-SRF gel (drug-containing plain gel).
Comment 6: Recent research should be added to the reference list, along with references to stability and comparable nano-delivery methods, as well as citations demonstrating the anti-arthritic properties of SRF and the synergy between MTX and SRF. To improve the manuscript, using the following reference is recommended: ”https://doi.org/10.1021/acsabm.5c00257”.
Add this in 2.3 of the manuscript
Response6:
Thanks for this valuable comment. We have updated our reference library and incorporated the stated study in our revised manuscript. Please refer to 2.3 of the revised manuscript.
Comment 7: The conclusion suggests further pharmacokinetic evaluation, but no preliminary data are provided. The authors need to provide initial pharmacokinetic data to support the claim of enhanced bioavailability. MTX and SRF are associated with significant toxicities, and any toxicity assessed in the in vivo study should be addressed.
Response 7:
We are grateful to the reviewer for this valuable input on our manuscript. In the conclusion, it was concluded that to excel our designed nanocarriers (MTX-SRF-TFSG), we need to evaluate it on the pharmacokinetic grounds, as due to limited resources and funding, we were not able to conduct ADME studies. Furthermore, the quantity of MTX and SRF administered to the mice through MTX-SRF-TFSG was less than the usual therapeutic doses.
Comment 8: To highlight its distinctive feature and advantages, the text would benefit from a comparison of the MTX-SRF-TFS formulation with other transdermal nano-delivery methods (such as liposomes and ethosomes) for the treatment of RA.
Response 8:
We are grateful to the reviewer for raising this important point. We have already incorporated the distinctive benefit of transferosomes into other lipid nanocarriers. Please refer to the introduction of the revised manuscript. (line 64-70)
The references are also updated.
References:
Dar, M. J., et al. (2020). "Development and evaluation of novel miltefosine-polyphenol co-loaded second generation nano-transfersomes for the topical treatment of cutaneous leishmaniasis." Expert opinion on drug delivery 17(1): 97-110.
Gözel, N., et al. (2017). "Sorafenib reveals anti-arthritic potentials in collagen induced experimental arthritis model." Archives of Rheumatology 33(3): 309.
Salem, H. F., et al. (2022). "Nano methotrexate versus methotrexate in targeting rheumatoid arthritis." Pharmaceuticals 16(1): 60.
Wang, Z. Z., et al. (2018). "Antiarthritic effects of sorafenib in rats with adjuvant‐induced arthritis." The Anatomical Record 301(9): 1519-1526.